# Association between Physical Activity and Phase Angle Obtained via Bioelectrical Impedance Analysis in South Korean Adults Stratified by Sex

**DOI:** 10.3390/nu16132136

**Published:** 2024-07-04

**Authors:** Jiwon Yang, Jiho Yu, Jinhyun Kim, Euncheol Park

**Affiliations:** 1Medical Courses, Yonsei University College of Medicine, Seoul 03722, Republic of Korea; jiwon.yang21@med.yuhs.ac (J.Y.); jiho.yu21@med.yuhs.ac (J.Y.); 2Department of Preventive Medicine, Yonsei University College of Medicine, Seoul 03722, Republic of Korea; 3Institute of Health Services Research, Yonsei University, Seoul 03722, Republic of Korea; 4Department of Psychiatry, Yonsei University College of Medicine, Seoul 03722, Republic of Korea

**Keywords:** physical activity, phase angle, sex, muscle-strengthening activity, leisure-time physical activity

## Abstract

This cross-sectional study aimed to examine the association of various aspects of physical activity, including intensity, duration, type, and purpose, with the phase angle (PhA), an objective indicator of health, in Korean adults after stratification by sex. Data from the 2022 Korean National Health and Nutrition Examination Survey, a nationwide, representative, population-based survey, were used. In total, 3996 participants were included in the study. Participants self-reported their weekly intensity, frequency, duration of engagement in physical activity. PhA was categorized into two groups on the basis of sex-specific averages. Multiple logistic regression analysis was used to investigate the relationship between physical activity and PhA, and proportional odds logistic regression analysis was performed to determine the association between physical activity and different subclasses of PhA. A positive association was found between sufficiently active aerobic physical activity and PhA compared with inactive physical activity (sufficiently active, male: odds ratio = 1.952, 95% confidence interval = 1.373–2.776; female: odds ratio = 1.333, 95% confidence interval = 1.019–1.745). This association was further strengthened when aerobic physical activity was accompanied by muscle-strengthening activity (sufficiently active with muscle-strengthening activity, male: aOR = 2.318, 95% CI = 1.512–3.554; female: aOR = 1.762, 95% CI = 1.215–2.556) and vigorous-intensity activities (sufficiently active with sufficient vigorous-intensity activity, male: aOR = 2.785, 95% CI = 1.647–4.709; female: aOR = 2.505, 95% CI = 1.441–4.356) and when there was more leisure-time physical activity than occupational physical activity (sufficiently active with more leisure-time physical activity, male: aOR = 2.158, 95% CI = 1.483–3.140; female: aOR = 1.457, 95% CI = 1.078–1.969). Furthermore, the inclusion of muscle-strengthening activity made a significant difference in the values of PhA for males with insufficiently active physical activity (aOR = 2.679, 95% CI = 1.560–4.602). For females with highly active physical activity (aOR = 1.521, 95% CI = 1.068–2.166), the inclusion of muscle-strengthening and vigorous-intensity activities were significantly associated with higher values for PhA. This study can be utilized to provide specific suggestions for better health programs and can change perception that only occupational physical activity is enough. This study also indicated that PhA can be used for personalized health assessments.

## 1. Introduction

As the global ageing population continues to increase dramatically [1,2], maintaining overall health and well-being has emerged as a focal point of contemporary research and public health efforts. Physical activity is commonly recognized as a powerful tool for achieving good health, among many lifestyle factors. Several studies have documented the profound effects of regular physical activity on general health [3], including its preventative effect on weight gain, obesity, chronic heart disease, and Type 2 diabetes mellitus [4]. Recent guidelines published by the World Health Organization encourage adults to engage in a minimum of 150–300 min of moderate-intensity physical activity or 75–150 min of vigorous-intensity physical activity per week, in addition to two or more days of muscle-strengthening exercises, to gain exercise-induced health benefits [5].

With the increasing attention to physical activity, advancements in bioelectrical impedance analysis (BIA) have introduced numerous indicators of body composition for assessing adults’ well-being, including the phase angle (PhA). BIA examines the body’s composition by directly measuring the body’s resistance and reactance [6]. PhA is derived from these measurements using the formula: $\tan^{-1}(\frac{reactance}{resistance})\times 180{}^{\circ}/\pi$ [7]. PhA is a valuable marker of cellular health and physiological status [8,9], with higher values of PhA reflecting greater cellularity, stronger cell membrane integrity, and improved cell function [10]. Previous studies have suggested that PhA declines with age [11,12] and is linked to numerous objective markers of a healthy status, such as nutritional status [9,13,14], muscle strength [15], aerobic capacity [16], total fitness age score [17], and even mortality [18]. Furthermore, many clinical trials have shown that PhA is a promising prognostic tool for various illnesses, such as inflammatory bowel and liver disease [19], COPD [20], and cancer [13,21]. PhA has been proven to be a more reliable tool than other BIA parameters, such as fat mass and fat-free mass, which could easily give incorrect estimates, depending on the state of hydration of the subject [22].

Understanding the interplay between modifiable lifestyle factors, such as physical activity and physiological markers, is imperative for developing exercise programs that enhance public well-being. While many studies have related general physical activity to health benefits [23,24,25], research focusing on the specific aspects of physical activity that can improve objectively obtained health markers is lacking. Moreover, there are studies that have indicated that leisure-time physical activity can lead to more health benefits than occupational physical activity can, such as reducing the risk of long-term sickness absence, but no study has yet shown how the different purposes of physical activity can influence the objective health markers. In this study, we aimed to examine the association of various aspects of physical activity, including intensity, duration, type, and purpose, with PhA, an objective indicator of health, in Korean adults after stratification by sex [7]. We hypothesized that a higher degree of physical activity, a markedly longer duration, stronger intensity, and inclusion of muscle-strengthening and leisure-time physical activities would be positively associated with higher PhA in Korean adults.

## 2. Materials and Methods

### 2.1. Study Population and Data

This study utilized data from the 2022 Korean National Health and Nutrition Examination Survey (KNHANES), conducted annually by the Korean Disease Control and Prevention Agency. KNHANES is a nationwide, representative, population-based, cross-sectional health and nutrition survey. The inclusion of examinations of PhA in the survey began in 2022; thus, our choice of data focused on this year. The recruitment process was randomized through a multistage probability sampling method. It was further stratified on the basis of geographic location. This method allowed for an accurate and generalizable representation of the health and nutrition estimates of the Korean population. Initially, out of the 6265 participants who took part in the 2022 survey, we investigated 5322 adults aged ≥19 years. Among these adult participants, we excluded 455 individuals due to missing information regarding physical activity. Additionally, data from a further 619 participants were omitted because they either did not undergo the BIA test or were unable to do so due to pregnancy or implantation of a pacemaker/defibrillator, which could interfere with the test. Of the remaining 4248 participants, we excluded 252 with missing values for any relevant survey questions on the covariate variables. Ultimately, this study included 3996 participants (1755 males and 2241 females). This is shown in Figure 1 for the visualization of the selection process for the study participants. 

### 2.2. Measures

#### 2.2.1. Phase Angle

PhA in degrees (°) was obtained using bioelectrical impedance spectroscopy (InBody 970; Biospace, Seoul, Republic of Korea). The InBody 970 is a multifrequency bioelectrical impedance analyzer that measures the body’s composition through impedance, which is the resistance generated by the body when a microcurrent flows through it [26]. Past studies have shown that measurements from InBody 970 showed good agreement with measurements of dual-energy X-ray absorptiometry, a standard tool for measuring muscle mass, proving that Inbody 970 can be a reliable method for assessing appendicular lean mass, fat-free mass, and the percentage of body fat [27]. The participants were instructed to remain standing while the measurements were performed at a single frequency of 50 kHz on the right side of the body. Fasting and temperance before conducting BIA was recommended to the participants in order to obtain the most accurate PhA results. However, these conditions may have not been controlled before conducting the test, so a small margin of error in the values of PhA may exist [28]. Similar to previous studies, we classified the participants into two groups (below average and above average) according to the sex-specific averages of PhA results (5.77° for males and 4.88° for females) [18,29]. Calculating the average separately for each sex is important, because the findings from previous research have shown that the difference in PhA between males and females is statistically significant. Although there are reference values for specific diseases such as sarcopenic obesity [30], there is no standard value for classifying the health of the general population; thus, future studies should use population-specific reference values for PhA [7,10]. 

#### 2.2.2. Physical Activity

The level of physical activity was assessed using the Global Physical Activity Questionnaire. Participants were asked to self-report the total minutes, hours, and days spent per week engaging in moderate- and vigorous-intensity aerobic physical activities for leisure time and work. Each participant reported the frequency of muscle-strengthening activity per week separately from aerobic physical activity. Hence, in this study, total physical activity refers to aerobic activity, excluding muscle-strengthening activities. Moderate-intensity physical activity was defined as an activity sufficient to make the participant slightly breathless yet still capable of maintaining a conversation. Vigorous-intensity physical activity was defined as an activity that caused the participant to become extremely breathless and experience a rapid heart rate [31]. After converting each duration into minutes, we calculated the total physical activity for each participant by converting moderate- and vigorous-intensity physical activities into multiples of the metabolic equivalent of task (MET), which describes the energy expenditure. Each minute of vigorous-intensity physical activity was equivalent to 8.0 METs, while moderate-intensity physical activity was equivalent to 4.0 METs [32]. According to the 2018 Physical Activity Guidelines for Americans, adults are recommended to engage in at least 150 min of moderate-intensity physical activity, 75 min of vigorous-intensity physical activity, or 600 MET-min of physical activity per week [3]. The participants were categorized into three groups according to their total physical activity, measured in MET-min/week: inactive if their MET-min/week was 0, insufficiently active if it was below the recommended level of 600 MET-min/week, and sufficiently active if it was above 600 MET-min/week. 

We reclassified the participants on the basis of their total physical activity levels for analyzing the subgroups. Categories included inactive (0 MET-min/week), insufficiently active (0–600 MET-min/week), sufficiently active (600–1200 MET-min/week), and highly active (>1200 MET-min/week). We further divided the insufficiently active and sufficiently active physical activity groups into different subgroups on the basis of any accompanying muscle-strengthening activity, the ratio of leisure-time physical activity to occupational physical activity, and the sufficiency of vigorous-intensity activity (whether vigorous-intensity activity alone exceeded 600 MET-min/week).

#### 2.2.3. Covariates

Age, sex, education level, region of residence, marital status, income level, and employment status were included as sociodemographic covariates. These sociodemographic covariates were assessed because past research has shown that these factors can have a significant influence on one’s health [33,34,35,36,37]. Body mass index (BMI), sleep duration, alcohol status, smoking status, and comorbidities such as diabetes mellitus, high blood pressure, asthma, and kidney diseases were included as health-related covariates. Among these variables, BMI, educational level, alcohol status, smoking status, marital status, employment status, sleep duration, asthma, and kidney disease were self-reported. Diabetes and high blood pressure were determined using blood tests for each participant. 

The participants were classified as underweight, normal, overweight, or obese on the basis of their BMI: <18.5 kg/m^2^, between 18.5 kg/m^2^ and 23 kg/m^2^, between 23 kg/m^2^ and 25 kg/m^2^, and >25 kg/m^2^, respectively. Educational level was divided into three groups: lower than middle school, high school, and college or above. Alcohol status was categorized into three groups: non-drinkers (participants who did not drink at all), social drinkers (participants who drank once or twice a month), and current drinkers (participants who drank every week). Smoking status was categorized into two groups: ever-smokers and non-smokers. The residential region was divided into three groups: metropolitan, urban, and rural. Income levels were categorized as low, middle-low, middle-high, and high on the basis of the income percentile of the participants. Participants whose income percentile fell within the <25th percentile were classified as having a low income; those falling between the >25th and <50th percentiles were categorized as having a middle-low income; those falling between the >50th and <75th percentiles were categorized as having a middle-high income; those in the >75th percentile were classified as having a high income. Sleep duration was divided into four groups based on a cutoff of 6 h, rounded to the nearest hour of the participants’ average sleep duration, and 8 h, the recommended sleep duration for adults [38]. Diabetes, high blood pressure, asthma, and kidney disease were classified into two groups depending on whether the participants had these conditions at the time of the survey. 

#### 2.2.4. Statistical Analysis

All analyses in the present study were stratified according to sex. Chi-square tests were used to analyze the participants’ characteristics. We investigated the link between physical activity and PhA using multiple logistic regression with adjustments for the covariates. We conducted subgroup analyses based on the previously mentioned covariates to examine whether these variables could potentially moderate the observed effects and to confirm that the results remained consistent across various subgroups. Additionally, subgroup analyses were performed to explore how different aspects of physical activity affected measurements of PhA. We also conducted a proportional odds logistic regression to investigate the association between physical activity and the three subgroups of PhA (below average, and two subgroups of above-average PhA, split according to the median). Additional analyses, including sensitivity analyses, were performed via binary and multinomial logistic regression, adopting different cutoffs for PhA (reference values for a healthy Hispanic population [39], and the age-specific mean PhA), incorporating different subgroups of PhA (divided into quartiles), and using different methods to group the PhA (divided by quartiles instead of averages).

All results are presented as the adjusted odds ratio (aOR) with 95% confidence intervals (CIs) to depict the strength and significance of the associations. Stratified sampling and weighted variables were used to account for potential biases. Analyses were conducted using SAS software (version 9.4; SAS Institute, Cary, NC, USA). Statistical significance was set at *p* < 0.053. We confirmed that multicollinearity did not exist in the study because all the variance inflation factors were less than 2.03.

## 3. Results

The study population was stratified by sex, and the general characteristics are presented in Table 1. In total, 3996 participants, comprising 1755 males (43.9%) and 2241 females (56.1%), were included in the analysis. Among males, 53.3% had above-average PhA, while the percentage was 53.2% among females. The group of participants with sufficiently active physical activity reported a higher percentage of above-average PhA. In particular, 42.4% of males in the inactive group, 58.4% in the insufficiently active group, and 68.0% in the sufficiently active group reported above-average PhA. Additionally, 47.4% of female participants in the inactive group, 59.7% in the insufficiently active physical activity group, and 63.6% in the sufficiently active physical activity group reported above-average PhA.

Table 2 shows the results of the multiple logistic regression analysis of the relationship between above-average PhA and the degree of physical activity using covariates. Relative to the inactive group, which was the reference group, male participants in both the insufficiently active and sufficiently active groups of physical activity were more likely to have an above-average PhA (aOR = 1.540, 95% CI = 1.051–2.256 in the insufficiently active physical activity group; aOR = 1.952, 95% CI = 1.373–2.776 in the sufficiently active physical activity group). However, only the sufficiently active physical activity group was statistically significant among female participants (aOR = 1.333, 95% CI = 1.019–1.745). 

The results of the subgroup analyses are presented in Table 3 and Appendix A. After dividing the participants according to their degree of physical activity, most subgroups showed trends similar to those of the main results. In particular, among males aged 29–39, the overweight subgroup, the subgroup that sleeps for approximately 6–8 h, and subgroups without comorbidities, such as diabetes and high blood pressure, showed significantly higher odds of having an above-average PhA in both the insufficiently active and sufficiently active physical activity groups than those in the inactive group. In addition, among males, the age group of 40–49, subgroups with a normal weight, and the subgroups that sleep for under 6 h or 8–10 h had significantly higher odds of having above-average PhA in the sufficiently active physical activity group than in the inactive group. Among females, participants without diabetes were more likely to have above-average PhA when engaging in sufficiently active physical activity. 

Subgroup analyses of physical activity are shown in Table 4. Generally, males who engaged in physical activity showed significantly higher odds of having above-average PhA than inactive participants. However, the subgroup with insufficiently active with no muscle-strengthening activity and subgroups that engaged more in occupational physical activity than leisure-time physical activity were the exceptions. Among females, positive associations with a higher PhA were only observed in the subgroups with highly active physical activity (aOR = 1.521, 95% CI = 1.068–2.166), sufficiently active physical activity with muscle-strengthening activity (aOR = 1.762, 95% CI = 1.215–2.556), sufficiently active physical activity and engaging in more leisure-time physical activity (aOR = 1.457, 95% CI = 1.078–1.969), and sufficiently active physical activity with sufficient vigorous-intensity activity (aOR = 2.505, 95% CI = 1.441–4.356).

Figure 2 displays the results of the proportional odds logistic regression, representing the analysis of the association between physical activity and the three subgroups of the PhA (below average, and the two subgroups of above-average PhA split on the basis of the median). It showed a similar trend to the main results. Among males, there was a statistically significant association where being sufficiently active (aOR = 1.833, 95% CI = 1.409–2.384) or insufficiently active (aOR = 1.535, 95% CI = 1.075–2.193) increased the odds of higher PhA categories compared with being inactive. For females, the odds ratio of 1.131 (95% CI: 0.844–1.515) indicated a non-significant trend towards a higher odds of higher PhA categories for individuals who were insufficiently active compared with those who were inactive, though this effect was not statistically significant at the conventional 5% level. When comparing females who were sufficiently active to those who were inactive, the odds of having a higher PhA were 1.380 times higher for those who were sufficiently active compared with those who were inactive.

## 4. Discussion

This study aimed to identify the association of various aspects of physical activity, including intensity, duration, type, and purpose, with the PhA. Our results have shown that a higher degree of physical activity, especially activity with a longer duration, stronger intensity, the inclusion of muscle-strengthening activities, and engagement in leisure-time physical activity, was positively associated with a higher PhA among Korean adults. According to a systematic review and meta-analysis [23], exercise training effectively improves the body’s composition in postmenopausal women. There was also a study revealing high correlations between the lack of exercise and health disorders such as diabetes [40]. In particular, resistance training helps increase muscle mass, and aerobic training helps decrease fat mass. A study [24] revealed that 12 weeks of muscle-strengthening activity, especially resistance training, increased muscle mass, muscle strength, and physical function in older adults. There are also studies that have addressed both patterns of physical activity and dietary behaviors that are significant for the composition of body mass. Two linked studies [41,42] supported a strong association of physical activity patterns (leisure time, sitting time) and dietary behaviors (healthy eating, drinking carbonated drinks, and eating fast food), with sex differences. A study about metabolic syndrome showed that metabolic syndrome was strongly associated with groups that were characterized by a high intake of fast food and sweetened beverages, and low to moderate physical activity [43]. A study using the concept of joint temporal dietary and physical activity patterns [44] showed that clusters with higher physical activity were associated with more favorable health indicators. A previous study [8] demonstrated that daily moderate-to-vigorous intensity physical activity, step count, and exercise habits were significantly associated with the PhA in adults but failed to compare the effects of different types and intensities of physical activity on the PhA. A more recent study [25] focused on the association between three different intensities of physical activity and PhA in a limited sample of adults older than 65 years. However, it examined neither the correlation between diverse types of physical activity and PhA, nor the effect of different purposes of physical activity on the PhA. In our study, physical activity was further divided according to the following three categories: intensity (moderate intensity or vigorous intensity), purpose (occupational or leisure-time), and type (with or without muscle-strengthening activity).

Our results generally agreed with those of previous studies, which have shown a positive association between overall physical activity and PhA in older adults [25], and further supported the notion that individuals who exercise regularly have significantly higher values of PhA than those in the inactive group [8]. Moreover, the odds of being in the categories with higher PhA gradually increased with increased physical activity, indicating the requirement for more physical activity. In addition, the inclusion of muscle-strengthening activity with aerobic physical activity caused a significant difference in the values of PhA, further supporting our hypothesis. Muscle-strengthening activity was necessary to obtain a higher PhA for males engaged in insufficiently active physical activity. However, males who engaged in sufficiently active physical activity could achieve a higher PhA, regardless of muscle-strengthening activity. Since the odds were higher for muscle-strengthening activity, muscle-strengthening activity should be recommended in either case. For females, engaging in sufficiently active physical activity, in addition to muscle-strengthening activity, was necessary to achieve a higher PhA. Further research is needed to understand the mechanism underlying the association between high PhA and physical activity. Plausible explanations can be drawn from previous studies. Generally, healthy individuals typically have a PhA ranging from 5.0 to 7.0, whereas athletes engaging in extensive exercise and having considerable muscle mass may exceed 9.5 [17,45,46]. This variation partly explains why the PhA was significantly higher with greater physical activity and the incorporation of muscle-strengthening activity into the exercise regime.

For both males and females, a higher ratio of leisure-time physical activity to occupational physical activity was positively correlated with a higher PhA. However, females also needed to engage in sufficiently active physical activity to reap the benefits induced by physical activity. Leisure-time physical activity is more effective than occupational physical activity at reducing mortality [47] and metabolic syndrome [48]. Nonetheless, research directly related to PhA is lacking, necessitating further studies. 

In general, our results highlighted that females need to engage in a greater amount of physical activity than males to obtain the same health benefits from exercise in terms of PhA. Previous studies focusing on how sex influences the effects of exercise have shown conflicting results. One study reported that males experienced a greater loss of overall and abdominal fat after exercise, but the differences between the sexes were not statistically significant [49]. Another study reported that sex was not associated with the efficiency of exercise [50]. While studies have indicated that females gain greater health benefits, such as reductions in all-cause and cardiovascular mortality risks [51], than males from the same amount of leisure-time physical activity, other studies supported our results. According to a review article on the relationships among exercise, sex, and catecholamine [52], participants engaging in intense exercise had higher concentrations of catecholamine than untrained subjects. This phenomenon, termed the ‘sports adrenal medulla’, suggests that physical training can increase the adrenal glands’ volume and secretion of adrenaline. The same study indicated that females have a lower adrenaline response than males, suggesting that the effect of training is weaker in females. Another study found that gains in muscle mass from strength training in males were approximately double those in females [53]. Another study in older adults has shown that males have larger hypertrophic responses of the myofibers to identical resistance exercise than females [54]. We speculate that the rationale behind our results showing that females require more exercise may also be due to the differences in specific exercises each sex typically performs, and the difference in perception of what each sex regards as moderate and vigorous intensity. Therefore, a further study that delves deeper than just the type of exercise and focuses on the effect of specific exercises is needed to confirm whether differences exist in the degree of change in PhA depending on sex. Additionally, because there may be differences in not only the patterns of physical activity but also dietary behaviors depending on gender [41], follow-up research that considers both factors may be necessary. 

Furthermore, in males, a high PhA was especially significant in participants in their 30s and 40s (Appendix A), and it was significant in healthy individuals without comorbidities in general, regardless of sex. The decline in PhA with age may have occurred because a decline in aerobic capacity, commonly caused by old age, is strongly associated with the decreased efficiency of exercise [50]. Our results showed no significant differences between higher PhA and physical activity in participants with comorbidities. However, according to a review article identifying the effects of recreational soccer training on aerobic fitness and health in healthy individuals and clinical patients [55], recreational soccer, if practiced two or three times per week, yielded important health benefits for healthy and pathological individuals. This conflicting result in our study with previous studies may be due to the small sample size of individuals with comorbidities. Therefore, further studies on the association between physical activity and higher PhA in pathological individuals are needed when more years of data have been collected. For further analyses (Appendix A), we divided physical activity into quartiles for one analysis. We used the Hispanic reference value of the PhA [39,56] as our cutoff PhA for another analysis. Additionally, we subdivided the degree of muscle-strengthening activity further and even divided PhA into quartiles for sensitivity analysis. The results were consistent with our main findings.

This study had several limitations. First, as it was cross-sectional, we were unable to establish causality or the direction of the relationship between physical activity and PhA. Therefore, future longitudinal studies are required to draw cause-and-effect conclusions. Second, we used data from a self-reported survey (KNHANES). All physical activity data in the present study were self-reported, making them subjective and susceptible to recall bias. For instance, while leisure-time physical activity time may have been relatively accurate, occupational physical activity time may have been overestimated, as participants might have included long rest times in their total occupational physical activity time. Third, the survey measured the duration of muscle-strengthening exercises in days and aerobic exercises in minutes and hours, limiting a direct comparison between the two types. Consequently, detailed analyses of these exercises could not be performed using our results. Fourth, as we utilized the 2022 KNHANES data, this study’s generalizability to other races or cultures is limited. Finally, our study’s physical activity variable did not account for walking for transportation. Future studies should focus on subjects who can meet their MET requirements solely by walking to work. 

Further studies based on prospective designs are required to establish the mechanisms underlying this association, and research is needed to determine the types of physical activity that are associated with higher PhA in groups with specific diseases, such as high blood pressure or diabetes, and specific age groups, such as older people.

## 5. Conclusions

In conclusion, this study demonstrated a positive association between high PhA and physical activity. This association is strengthened when physical activity is accompanied by muscle-strengthening and vigorous-intensity activities. and when there is more leisure-time physical activity than occupational physical activity. This study can be utilized to provide specific suggestions for better health programs, personalized exercise prescriptions, and rehabilitation programs. This may be of particular help to females who must meet more demanding exercise conditions. In addition, since bioelectrical impedance spectroscopy is highly accessible in South Korea, people can easily perform personalized health assessments and check the effectiveness of their exercise routine by monitoring their PhA, a biophysical parameter that is measurable by BIA, and changes over time. Lastly, it may bring about a reconsideration of the perception of people who think they perform enough exercise through occupational physical activities.

## Figures and Tables

**Figure 1 nutrients-16-02136-f001:**
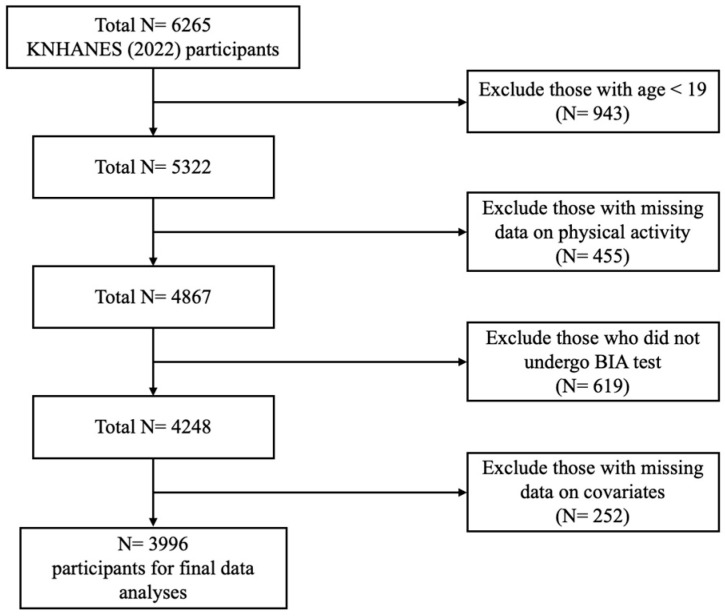
Flowchart for selection of the study participants. Data are from the 2022 Korean National Health and Nutrients Examination Survey (KNHANES).

**Figure 2 nutrients-16-02136-f002:**
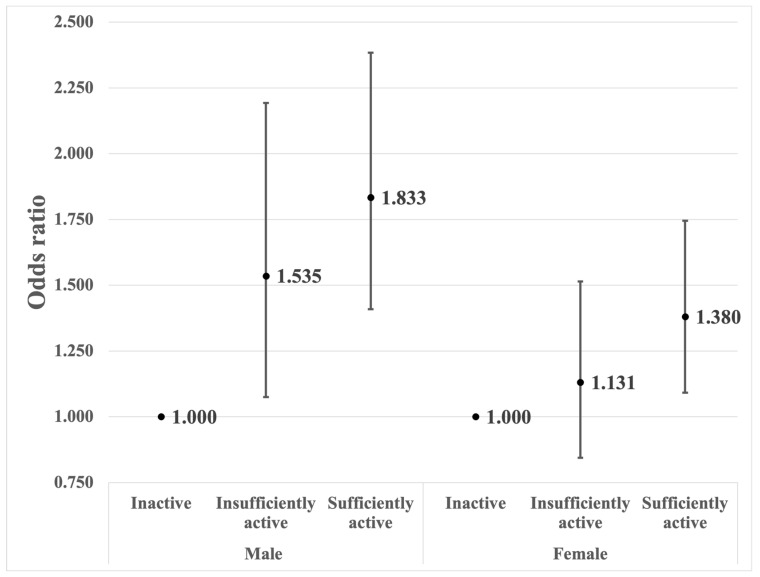
Associations between the amount of physical activity and the three subgroups of PhA (below average ^a^, and the two subgroups of above-average ^a^ PhA split on the basis of the median ^a^). Adjusted for age, body mass index, educational level, alcohol status, smoking status, region of residence, marital status, income level, employment status, sleep duration, and the presence of diabetes, high blood pressure, asthma, and kidney disease. ^a^ Average PhA: 5.77° for males and 4.88° for females; median value of above-average PhA: 6.2° for males and 5.2° for females.

**Table 1 nutrients-16-02136-t001:** Socioeconomic and health-related characteristics of all study participants according to PhA.

Variables	PhA													
	Male (N = 1755)							Female (N = 2241)						
	Total		Above Average ^a^		Below Average ^a^		*p*-Value	Total		Above Average ^a^		Below Average ^a^		*p*-Value
	N	(%)	N	(%)	N	(%)		N	(%)	N	(%)	N	(%)	
**Physical activity ^b^**							<0.0001							<0.0001
Inactive	899	(51.2)	381	(42.4)	518	(57.6)		1353	(60.4)	641	(47.4)	712	(52.6)	
Insufficiently active	293	(16.7)	171	(58.4)	122	(41.6)		350	(15.6)	209	(59.7)	141	(40.3)	
Sufficiently active	563	(32.1)	383	(68.0)	180	(32.0)		538	(24.0)	342	(63.6)	196	(36.4)	
**Age**							<0.0001							<0.0001
19–28	218	(12.4)	165	(75.7)	53	(24.3)		257	(11.5)	138	(53.7)	119	(46.3)	
29–39	252	(14.4)	201	(79.8)	51	(20.2)		310	(13.8)	207	(66.8)	103	(33.2)	
40–49	283	(16.1)	211	(74.6)	72	(25.4)		406	(18.1)	265	(65.3)	141	(34.7)	
50–59	299	(17.0)	199	(66.6)	100	(33.4)		414	(18.5)	275	(66.4)	139	(33.6)	
60–69	379	(21.6)	133	(35.1)	246	(64.9)		487	(21.7)	245	(50.3)	242	(49.7)	
70+	324	(18.5)	26	(8.0)	298	(92.0)		367	(16.4)	62	(16.9)	305	(83.1)	
**Body mass index ^c^**							<0.0001							<0.0001
Underweight	41	(2.3)	7	(17.1)	34	(82.9)		127	(5.7)	41	(32.3)	86	(67.7)	
Normal weight	532	(30.3)	185	(34.8)	347	(65.2)		964	(43.0)	441	(45.7)	523	(54.3)	
Overweight	412	(23.5)	228	(55.3)	184	(44.7)		491	(21.9)	289	(58.9)	202	(41.1)	
Obese	770	(43.9)	515	(66.9)	255	(33.1)		659	(29.4)	421	(63.9)	238	(36.1)	
**Education level**							<0.0001							<0.0001
Lower than middle school	350	(19.9)	92	(26.3)	258	(73.7)		605	(27.0)	222	(36.7)	383	(63.3)	
High school	612	(34.9)	338	(55.2)	274	(44.8)		741	(33.1)	433	(58.4)	308	(41.6)	
College or above	793	(45.2)	505	(63.7)	288	(36.3)		895	(39.9)	537	(60.0)	358	(40.0)	
**Alcohol status ^d^**							<0.0001							<0.0001
Non-drinker	597	(34.0)	252	(42.2)	345	(57.8)		1334	(59.5)	636	(47.7)	698	(52.3)	
Social drinker	593	(33.8)	344	(58.0)	249	(42.0)		637	(28.4)	386	(60.6)	251	(39.4)	
Current drinker	565	(32.2)	339	(60.0)	226	(40.0)		270	(12.0)	170	(63.0)	100	(37.0)	
**Smoking status**							<0.0001							0.0050
Ever-smoker	508	(28.9)	311	(61.2)	197	(38.8)		104	(4.6)	67	(64.4)	37	(35.6)	
Non-smoker	1247	(71.1)	624	(50.0)	623	(50.0)		2137	(95.4)	1125	(52.6)	1012	(47.4)	
**Region of residence**							0.5787							0.4775
Metropolitan	745	(42.5)	399	(53.6)	346	(46.4)		988	(44.1)	513	(51.9)	475	(48.1)	
Urban	660	(37.6)	358	(54.2)	302	(45.8)		843	(37.6)	452	(53.6)	391	(46.4)	
Rural	350	(19.9)	178	(50.9)	172	(49.1)		410	(18.3)	227	(55.4)	183	(44.6)	
**Marital status**							<0.0001							<0.0001
Married	1213	(69.1)	583	(48.1)	630	(51.9)		1435	(64.0)	827	(57.6)	608	(42.4)	
Single	542	(30.9)	352	(64.9)	190	(35.1)		806	(36.0)	365	(45.3)	441	(54.7)	
**Income level ^e^**							0.5767							0.9105
Low	419	(23.9)	212	(50.6)	207	(49.4)		538	(24.0)	292	(54.3)	246	(45.7)	
Middle-low	426	(24.3)	234	(54.9)	192	(45.1)		553	(24.7)	288	(52.1)	265	(47.9)	
Middle-high	453	(25.8)	247	(54.5)	206	(45.5)		581	(25.9)	310	(53.4)	271	(46.6)	
High	457	(26.0)	242	(53.0)	215	(47.0)		569	(25.4)	302	(53.1)	267	(46.9)	
**Employment status**							<0.0001							<0.0001
Employed	1250	(71.2)	781	(62.5)	469	(37.5)		1256	(56.0)	732	(58.3)	524	(41.7)	
Unemployed	505	(28.8)	154	(30.5)	351	(69.5)		985	(44.0)	460	(46.7)	525	(53.3)	
**Sleep duration**							<0.0001							0.0003
<6 h	215	(12.3)	98	(45.6)	117	(54.4)		364	(16.2)	160	(44.0)	204	(56.0)	
≥6 h and <8 h	1043	(59.4)	604	(57.9)	439	(42.1)		1277	(57.0)	702	(55.0)	575	(45.0)	
≥8 h and <10 h	469	(26.7)	224	(47.8)	245	(52.2)		564	(25.2)	316	(56.0)	248	(44.0)	
≥10 h	28	(1.6)	9	(32.1)	19	(67.9)		36	(1.6)	14	(38.9)	22	(61.1)	
**Diabetes**							<0.0001							<0.0001
No	1461	(83.2)	828	(56.7)	633	(43.3)		1989	(88.8)	1089	(54.8)	900	(45.2)	
Yes	294	(16.8)	107	(36.4)	187	(63.6)		252	(11.2)	103	(40.9)	149	(59.1)	
**High blood pressure**							<0.0001							<0.0001
No	1108	(63.1)	682	(61.6)	426	(38.4)		1624	(72.5)	924	(56.9)	700	(43.1)	
Yes	647	(36.9)	253	(39.1)	394	(60.9)		617	(27.5)	268	(43.4)	349	(56.6)	
**Asthma**							0.0245							0.0566
No	1704	(97.1)	914	(53.6)	790	(46.4)		2172	(96.9)	1151	(53.0)	1021	(47.0)	
Yes	51	(2.9)	21	(41.2)	30	(58.8)		69	(3.1)	41	(59.4)	28	(40.6)	
**Kidney disease**							0.0208							0.0900
No	1722	(98.1)	923	(53.6)	799	(46.4)		2195	(97.9)	1165	(53.1)	1030	(46.9)	
Yes	33	(1.9)	12	(36.4)	21	(63.6)		46	(2.1)	27	(58.7)	19	(41.3)	
**Total**	1755	(100.0)	935	(53.3)	820	(46.7)		2241	(100.0)	1192	(53.2)	1049	(46.8)	

^a^ Average PhA: 5.77° for males and 4.88° for females. ^b^ Divided according to energy expenditure (multiplicity of 4.0 METs for moderate-intensity physical activity, 8.0 METs for vigorous-intensity physical activity). Participants were classed as ‘inactive’ at 0 MET-min/week, ‘insufficiently active’ at <600 MET-min/week, and ‘sufficiently active’ at >600 MET-min/week. ^c^ ‘Underweight’: <18.5 kg/m^2^; ‘normal weight’: between 18.5 kg/m^2^ and 23.0 kg/m^2^; ‘overweight’: between 23.0 kg/m^2^ and 25.0 kg/m^2^; ‘obese’: >25.0 kg/m^2. d^ ‘Non-drinker’: not drinking at all; ‘social drinker’: drinking once or twice every month; ‘current drinker’: drinking every week. ^e^ Categorised into ‘low’, ‘middle low’, ‘middle high’, and ‘high’ if the income percentile of the participant was in the <25th percentile, >25th percentile and <50th percentile, >50th percentile and <75th percentile, and >75th percentile, respectively.

**Table 2 nutrients-16-02136-t002:** Association between the amount of physical activity and above-average PhA.

Variables	Male (N = 1755)				Female (N = 2241)			
	Above-Average PhA ^a^				Above-Average PhA ^a^			
	aOR ^b^	95% CI			aOR ^b^	95% CI		
**Physical activity ^c^**								
Inactive	1.000				1.000			
Insufficiently active	1.540	1.051	–	2.256	1.071	0.774	–	1.483
Sufficiently active	1.952	1.373	–	2.776	1.333	1.019	–	1.745
**Age**								
19–28	1.000				1.000			
29–39	0.868	0.484	–	1.559	0.982	0.607	–	1.588
40–49	0.479	0.254	–	0.903	0.842	0.537	–	1.321
50–59	0.387	0.197	–	0.760	0.805	0.506	–	1.279
60–69	0.117	0.059	–	0.231	0.343	0.209	–	0.564
70+	0.021	0.010	–	0.045	0.066	0.036	–	0.120
**Body mass index ^d^**								
Underweight	0.253	0.086	–	0.747	0.490	0.310	–	0.775
Normal weight	1.000				1.000			
Overweight	2.811	1.989	–	3.973	2.483	1.821	–	3.386
Obese	4.732	3.208	–	6.980	3.798	2.834	–	5.090
**Education level**								
Lower than middle school	1.504	0.907	–	2.495	1.149	0.818	–	1.615
High school	1.096	0.780	–	1.539	1.102	0.840	–	1.447
College or above	1.000				1.000			
**Alcohol status ^e^**								
Non-drinker	1.000				1.000			
Social drinker	1.216	0.846	–	1.748	1.069	0.844	–	1.352
Current drinker	1.387	0.973	–	1.977	1.051	0.747	–	1.477
**Smoking status**								
Ever-smoker	1.404	1.054	–	1.870	1.667	1.051	–	2.643
Non-smoker	1.000				1.000			
**Region of residence**								
Metropolitan	1.000				1.000			
Urban	1.136	0.795	–	1.624	0.937	0.735	–	1.194
Rural	2.144	1.439	–	3.195	1.304	0.956	–	1.778
**Marital status**								
Married	1.000				1.000			
Single	1.087	0.724	–	1.632	0.627	0.467	–	0.843
**Income level ^f^**								
Low	1.000				1.000			
Middle low	1.173	0.736	–	1.868	0.877	0.655	–	1.173
Middle high	1.085	0.730	–	1.613	1.007	0.732	–	1.385
High	1.276	0.791	–	2.058	1.026	0.755	–	1.394
**Employment status**								
Employed	1.000				1.000			
Unemployed	0.572	0.387	–	0.844	0.829	0.657	–	1.046
**Sleep duration**								
<6 h	0.816	0.540	–	1.234	0.967	0.707	–	1.323
≥6 h and <8 h	1.000				1.000			
≥8 h and <10 h	0.861	0.614	–	1.208	1.286	0.991	–	1.667
≥10 h	0.787	0.341	–	1.817	0.462	0.212	–	1.004
**Diabetes**								
No	1.000				1.000			
Yes	0.562	0.378	–	0.836	0.712	0.474	–	1.068
**High blood pressure**								
No	1.000				1.000			
Yes	0.713	0.483	–	1.052	1.028	0.743	–	1.422
**Asthma**								
No	1.000				1.000			
Yes	0.318	0.145	–	0.699	1.249	0.666	–	2.341
**Kidney disease**								
No	1.000				1.000			
Yes	1.073	0.381	–	3.019	1.012	0.529	–	1.936

Abbreviations: aOR, adjusted odds ratio; CI, confidence interval. ^a^ Average PhA: 5.77° for males and 4.88° for females. ^b^ Adjusted for the amount of physical activity, age, body mass index, educational level, alcohol status, smoking status, region of residence, marital status, income level, employment status, sleep duration, and the presence of diabetes, high blood pressure, asthma, and kidney disease. ^c^ Divided on the basis of energy expenditure (multiplicity of 4.0 METs for moderate-intensity physical activity, 8.0 METs for vigorous-intensity physical activity). Participants were classed as ‘inactive’ at 0 MET-min/week, ‘insufficiently active’ at <600 MET-min/week, and ‘sufficiently active’ at >600 MET-min/week. ^d^ ‘Underweight’: <18.5 kg/m^2^; ‘normal weight’: between 18.5 kg/m^2^ and 23.0 kg/m^2^; ‘overweight’: between 23.0 kg/m^2^ and 25.0 kg/m^2^; ‘obese’: >25.0 kg/m^2^. ^e^ ‘Non-drinker’: not drinking at all; ‘social drinker’: drinking once or twice every month; ‘current drinker’: drinking every week. ^f^ Categorized into ‘low’, ‘middle low’, ‘middle high’, and ‘high’ if the income percentile of the participant was in the <25th percentile, >25th percentile and <50th percentile, >50th percentile and <75th percentile, and >75th percentile, respectively.

**Table 3 nutrients-16-02136-t003:** Results of the subgroup analysis stratified by the independent variables.

Variables	Inactive	Insufficiently Active				Sufficiently Active			
		Above-Average PhA ^a^				Above-Average PhA ^a^			
	aOR ^b^	aOR ^b^	95% CI			aOR ^b^	95% CI		
**Males (N = 1755)**									
**Age**									
19–28	1.000	2.188	0.624	–	7.666	1.594	0.633	–	4.012
29–39	1.000	4.862	1.195	–	19.786	6.136	2.030	–	18.546
40–49	1.000	1.657	0.638	–	4.300	3.983	1.749	–	9.071
50–59	1.000	0.832	0.350	–	1.975	1.031	0.495	–	2.149
60–69	1.000	1.706	0.812	–	3.583	1.789	0.879	–	3.642
70+	1.000	2.923	0.849	–	10.070	1.084	0.330	–	3.564
**Body mass index ^c^**									
Underweight	1.000	N.A	N.A	–	N.A	N.A	N.A	–	N.A
Normal weight	1.000	1.431	0.732	–	2.795	3.583	2.106	–	6.096
Overweight	1.000	2.622	1.076	–	6.388	2.471	1.124	–	5.433
Obese	1.000	1.598	0.910	–	2.804	1.463	0.825	–	2.594
**Sleep duration**									
<6 h	1.000	2.589	0.640	–	10.480	7.393	2.550	–	21.434
≥6 h and <8 h	1.000	1.597	1.007	–	2.530	1.878	1.233	–	2.859
≥8 h and <10 h	1.000	1.248	0.492	–	3.165	1.899	1.063	–	3.393
≥10 h	1.000	N.A	N.A	–	N.A	3.072	0.186	–	50.648
**Diabetes**									
No	1.000	1.648	1.087	–	2.497	2.124	1.449	–	3.113
Yes	1.000	1.160	0.441	–	3.050	1.175	0.465	–	2.968
**High blood pressure**									
No	1.000	1.818	1.133	–	2.920	2.585	1.693	–	3.947
Yes	1.000	1.216	0.600	–	2.467	0.933	0.517	–	1.682
**Females (N = 2241)**									
**Age**									
19–28	1.000	2.172	0.776	–	6.078	2.809	1.123	–	7.027
29–39	1.000	1.460	0.664	–	3.214	1.814	0.938	–	3.508
40–49	1.000	0.741	0.381	–	1.441	0.820	0.450	–	1.494
50–59	1.000	1.359	0.675	–	2.736	1.300	0.660	–	2.561
60–69	1.000	0.917	0.405	–	2.072	1.843	1.066	–	3.187
70+	1.000	0.546	0.128	–	2.323	0.627	0.144	–	2.719
**Body mass index ^c^**									
Underweight	1.000	0.664	0.144	–	3.061	1.336	0.470	–	3.795
Normal weight	1.000	1.361	0.892	–	2.077	1.396	0.947	–	2.059
Overweight	1.000	0.778	0.382	–	1.585	1.793	0.960	–	3.346
Obese	1.000	0.865	0.427	–	1.752	0.908	0.530	–	1.554
**Sleep duration**									
<6 h	1.000	0.543	0.180	–	1.640	1.639	0.775	–	3.466
≥6 h and <8 h	1.000	1.077	0.744	–	1.559	1.142	0.809	–	1.610
≥8 h and <10 h	1.000	1.197	0.672	–	2.135	1.641	0.889	–	3.031
≥10 h	1.000	N.A	N.A	–	N.A	N.A	N.A	–	N.A
**Diabetes**									
No	1.000	1.080	0.770	–	1.515	1.432	1.090	–	1.882
Yes	1.000	1.442	0.297	–	6.996	0.619	0.272	–	1.412
**High blood pressure**									
No	1.000	1.099	0.757	–	1.595	1.330	0.981	–	1.805
Yes	1.000	1.078	0.527	–	2.208	1.269	0.695	–	2.318

aOR, adjusted odds ratio; CI, confidence interval; N.A., not applicable. More details are in Appendix A. N.A. is due to a small sample size. ^a^ Average PhA: 5.77° for males and 4.88° for females. ^b^ Adjusted for age, body mass index, educational level, alcohol status, smoking status, region of residence, marital status, income level, employment status, sleep duration, and the presence of diabetes, high blood pressure, asthma, and kidney disease. ^c^ ‘Underweight’: <18.5 kg/m^2^; ‘normal weight’: between 18.5 kg/m^2^ and 23.0 kg/m^2^; ‘overweight’: between 23.0 kg/m^2^ and 25.0 kg/m^2^; ‘obese’: >25.0 kg/m^2^.

**Table 4 nutrients-16-02136-t004:** Analysis of the association between specific types of physical activity and above-average PhA.

Variables			Male				Female			
			Above-Average PhA ^a^				Above-Average PhA ^a^			
			aOR ^b^	95% CI			aOR ^b^	95% CI		
**Physical activity ^c^**										
	Inactive		1.000				1.000			
		Insufficiently active	1.538	1.052	–	2.251	1.070	0.775	–	1.479
		Sufficiently active	1.763	1.087	–	2.860	1.169	0.838	–	1.630
		Highly active	2.071	1.409	–	3.043	1.521	1.068		2.166
		Insufficiently active with no muscle-strengthening activity	1.061	0.654	–	1.722	0.956	0.669	–	1.366
		Sufficiently active with no muscle-strengthening activity	1.616	1.032	–	2.530	1.127	0.790	–	1.606
		Insufficiently active with muscle-strengthening activity	2.679	1.560	–	4.602	1.530	0.866	–	2.702
		Sufficiently active with muscle-strengthening activity	2.318	1.512	–	3.554	1.762	1.215	–	2.556
		Insufficiently active and more occupational physical activity	1.127	0.570	–	2.227	1.144	0.546	–	2.399
		Sufficiently active and more occupational physical activity	1.529	0.883	–	2.648	1.029	0.679	–	1.560
		Insufficiently active and more leisure-time physical activity	1.694	1.123	–	2.557	1.058	0.760	–	1.474
		Sufficiently active and more leisure-time physical activity	2.158	1.483	–	3.140	1.457	1.078	–	1.969
		Insufficiently active	1.537	1.051	–	2.247	1.082	0.784	–	1.494
		Sufficiently active with insufficient vigorous-intensity activity ^d^	1.627	1.095	–	2.417	1.069	0.811	–	1.410
		Sufficiently active with sufficient vigorous-intensity activity ^d^	2.785	1.647	–	4.709	2.505	1.441	–	4.356

Abbreviations: aOR, adjusted odds ratio; CI, confidence interval. ^a^ Average PhA: 5.77° for males and 4.88° for females. ^b^ Adjusted for age, body mass index, educational level, alcohol status, smoking status, region of residence, marital status, income level, employment status, sleep duration, and the presence of diabetes, high blood pressure, asthma, and kidney disease. ^c^ Divided on the basis of energy expenditure (multiplicity of 4.0 METs for moderate-intensity physical activity, 8.0 METs for vigorous-intensity physical activity). Participants were classed as ‘inactive’ at 0 MET-min/week, ‘insufficiently active’ at <600 MET-min/week, and ‘sufficiently active’ at >600 MET-min/week. ^d^ ‘Sufficiently active with insufficient vigorous-intensity activity’: total physical activity was >600 MET-min/week but vigorous-intensity activity alone was <600 MET-min/week; ‘sufficiently active with sufficient vigorous-intensity activity’: total physical activity was >600 MET-min/week and vigorous-intensity activity alone was >600 MET-min/week.

## Data Availability

The data analyzed in this study are publicly accessible on the KNHANES website, administered by the Korean Disease Control and Prevention Agency (https://knhanes.kdca.go.kr/knhanes/main.do, accessed on 8 April 2024).

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
