# Peer review of "Association between Physical Activity and Phase Angle Obtained via Bioelectrical Impedance Analysis in South Korean Adults Stratified by Sex"

_nutrients, 2024, doi:10.3390/nu16132136_

Round 1
Reviewer 1 Report
Comments and Suggestions for Authors
This study aimed to analyze the association between physical activity and phase angle in an adult population. While interesting I feel that a more proper rationale for the study in terms of the significance of phase angle and a more comprehensive approach in the statistical analysis, allowing a simplification of results report, should be considered.
-While “PA” is commonly used for physical activity, to prevent confusions I suggest using the whole term “phase angle” instead of the abbreviation.
-Line 17. I guess that not only duration of physical activity was reported.
-Lines 24-25 (and elsewhere). This sentence should be rewritten. It seems that, among others, muscle strengthening was not considered as physical activity. This is important and should be clarified here and elsewhere, even in the statistical analysis and the physical activity measurement, as the term physical activity per se includes all these exercises reported by the authors. When author report physical activity, does this parameter include all physical activity performed or only a specific fraction of it?
-Lines 24-29. Please, provide statistical values.
-Lines 49-62. While this reviewer appreciates all information provided, I feel that more information regarding, for example, a validation of phase angle values should be provided, e.g. studies reporting associations between phase angle and objective markers of a healthy status. This includes also the lack of well-stablished reference values as it is stated later in the manuscript.
-Lines70-73. This hypothesis has not been properly justified.
-Authors should clarify whether common limitations attributed to bioimpedance methodology for measurements of body composition are applicable to phase angle measurements.
-Line 105. Was any previously validated questionnaire, such as the IPAQ, used to determine al physical activity parameters?
-Line 132. Covariates. A rationale for the inclusion of most of these covariates has not been considered in the manuscript.
-Lines 139-155. Please explain smoking status classification. Furthermore, reconsider this classification taking into account other more commonly used (non-smokers, former smokers, occasional or social and smokers).
-Tables 3-4. In my opinion the application of a proper multilogistic regressions analysis would largely improve the quality of the manuscript and results reported. Otherwise, one could suggest that that most results suppose a repetition because covariates are probably sociated among themselves. This regression analysis would be able to consider differences between genders in a more comprehensive way.
-Lines 63-78. Please include the main results of the study within these initial lines of the discussion.
-Discussion. In my opinion the discussion should be rewritten as its content supposes mainly a repetition of results, even with continuous references to tables and figures.
-The ethics committee approval and guidelines followed should be reported.
Author Response
Comments 1: This study aimed to analyze the association between physical activity and phase angle in an adult population. While interesting I feel that a more proper rationale for the study in terms of the significance of phase angle and a more comprehensive approach in the statistical analysis, allowing a simplification of results report, should be considered.
Response 1: Thank you very much for the opportunity to revise our manuscript. We greatly appreciate the time and effort you have invested in evaluating our work, and we believe your suggestions have significantly improved the quality of our paper. Below, we address each of your comments in detail. We have carefully considered each comment and have revised the manuscript accordingly to improve its clarity and quality. Thank you again for your insightful feedback and for considering our revised manuscript. We sincerely hope that the revised manuscript meets your expectations.
Comments 2: While “PA” is commonly used for physical activity, to prevent confusions I suggest using the whole term “phase angle” instead of the abbreviation.
Response 2: Thank you for this suggestion. We agree using “PA” as an abbreviation for phase angle could cause confusion because “PA” is commonly used for abbreviating physical activity, which is another variable mentioned in the study. Therefore, we have replaced all “PA” with “phase angle” to prevent any confusion.
Comments 3: Line 17. I guess that not only duration of physical activity was reported.
Response 3: Thank you for pointing this out. It was indeed not only the duration of physical activity that was reported in the survey. Korean National Health and Nutrients Examination Survey reported intensity, frequency, and the duration of physical activity each participant has partaken. We have made additions to the manuscript to reflect this (page1, abstract, line 16-17). Specifically, we have replaced the sentence with the following: “Participants self-reported their weekly intensity, frequency, duration of physical activity engagement.”
Comments 4: Lines 24-25 (and elsewhere). This sentence should be rewritten. It seems that, among others, muscle strengthening was not considered as physical activity. This is important and should be clarified here and elsewhere, even in the statistical analysis and the physical activity measurement, as the term physical activity per se includes all these exercises reported by the authors. When author report physical activity, does this parameter include all physical activity performed or only a specific fraction of it?
Response 4: We apologize for the oversight. Upon review, we agree that our vague description of physical activity under the abstract section could potentially create ambiguity of whether muscle strengthening is considered as physical activity or not. We have clarified the scope of physical activity in our study by specifying “aerobic” when describing physical activity in the abstract section (page1, abstract, line21). Furthermore, under the methodology section(page3-4, physical activity section, line132-135), we clarify once again the scope of what we refer to as physical activity in this study by including the following sentences: “Each participant reported the frequency of muscle-strengthening activity per week separately from aerobic physical activity. Hence in this study, total physical activity refers to aerobic activity, excluding muscle-strengthening activities.”
Comments 5: Lines 24-29. Please, provide statistical values.
Response 5: Thank you for this feedback. We have revised the abstract section (page1, abstract, line25-33) to include the statistical values of our main results. Specifically, we have added the odds ratio and the confidence interval of our results.
Comments 6: Lines 49-62. While this reviewer appreciates all information provided, I feel that more information regarding, for example, a validation of phase angle values should be provided, e.g. studies reporting associations between phase angle and objective markers of a healthy status. This includes also the lack of well-stablished reference values as it is stated later in the manuscript.
Response 6: We appreciate your feedback regarding the phase angle variable. We agree that information regarding the validity of phase angle as an objective marker of health was lacking. Therefore, we have expanded this by adding a reference to a past study reporting the strong associations between phase angle and total fitness age score, which is based on standardized and comprehensive fitness assessments and provides multifaceted view of one’s physical health, in addition to studies reporting phase angle’s strong correlation to muscle strength, aerobic capacity, and mortality. This supports the validation of phase angle (page2, paragraph2, line63-64). Furthermore, we have expanded on the lack of well-established reference values of phase angle, specifically adding the following sentence under the phase angle section of methodology (page3, phase angle section, line 125-127): “Although there are reference values for specific diseases such as sarcopenic obesity, there is no standard value for classifying the health of the general population, thus, future studies should use population-specific reference values for phase angle.”
Comments 7: Lines70-73. This hypothesis has not been properly justified.
Response 7: Thank you for your comment. We agree that the reasoning for setting our hypothesis was lacking. Although we have mentioned the potential health benefits of high duration and frequency of physical activity and muscle strengthening activity prior to line 70, we have failed to justify why we think leisure-time physical activity could possibly improve health. Therefore, we added an explanation (page2, paragraph3, line74-78) about a previous study that indicates leisure-time physical activity’s potential to reduce risk of health disorders compared to occupational physical activity to display the thought process behind our hypothesis: “Moreover, there are studies that indicate leisure-time physical activity can lead to more health benefits than occupational physical activity can, such as reducing the risk for long-term sickness absence, but no study yet shows how the different purposes of physical activity can influence the objective health marker.”
Comments 8: Authors should clarify whether common limitations attributed to bioimpedance methodology for measurements of body composition are applicable to phase angle measurements.
Response 8: Thank you for this important suggestion. We have included a description about a possible limitation regarding bioimpedance methodology for measurements of body composition that may be applicable to phase angle measurements under the phase angle section of methodology (page3, phase angle section, line117-120). After the revision, we have acknowledged that slight errors in phase angle measurements could occur due to the participants not complying to the necessary conditions before conducting the bioelectrical impedance analysis. Specifically, we have included the following: “Fasting and temperance before conducting BIA is recommended to participants in order to obtain the most accurate phase angle results. However, these conditions may have not been controlled before conducting the test, so a small margin of error in phase angle values may exist.”
Comments 9: PAGE 3 Line 105. Was any previously validated questionnaire, such as the IPAQ, used to determine al physical activity parameters?
Response 9: Thank you for your comment. Global Physical Activity Questionnaire(GPAQ), a previously validated questionnaire similar to IPAQ, was used to determine the physical activity parameters. The source of our data, Korea National Health and Nutrition Examination Survey (KNHANES), used IPAQ until 2014, but since 2014, KNHANES have been using GPAQ developed by WHO to make up for the shortcomings of IPAQ.(reference: https://doi.org/10.1177/1757975919854301, https://doi.org/10.5763/kjsm.2020.38.1.1) To enhance clarification regarding this, we have included a description of the basis of physical activity parameters (page3, physical activity section, line130-131): “The level of physical activity was assessed using the Global Physical Activity Questionnaire”.
Comments 10: PAGE 3 Line 132. Covariates. A rationale for the inclusion of most of these covariates has not been considered in the manuscript.
Response 10: We greatly appreciate your feedback. We have made some revisions (page4, covariates section, line161-166) to the methodology section to provide more details on the possible influences of sociodemographic covariates and comorbidities on health in order to account for the inclusion of these covariates in our study. Specifically, we have included references to various past studies to support our reasoning behind including some sociodemographic factors that may appear to have a very weak relationship to health.
Comments 11: Lines 139-155. Please explain smoking status classification. Furthermore, reconsider this classification taking into account other more commonly used (non-smokers, former smokers, occasional or social and smokers).
Response 11: Thank you for bringing this to our attention. We have included the explanation of how we categorized smoking status (page4, covariates section, line176-177): “Smoking status was categorized into two groups: ever-smoker and non-smoker”. We used this categorization because the only information the source of our data, Korea National Health and Nutrition Examination Survey (KNHANES), provides about smoking is whether the participant has ever smoked or not.
Comments 12: Tables 3-4. In my opinion the application of a proper multilogistic regressions analysis would largely improve the quality of the manuscript and results reported. Otherwise, one could suggest that that most results suppose a repetition because covariates are probably sociated among themselves. This regression analysis would be able to consider differences between genders in a more comprehensive way.
Response 12: Thank you for your suggestion. We have revised the methodology section (page5, statistical analysis section, line200-201) of our manuscript to provide more details about the further multinomial logistic regression analyses that we had conducted where we divided phase angle into quartiles. Furthermore we have added a description of an additional analysis we had done where we used healthy male and female Hispanic reference values as the phase angle cutoff to explore the differences between genders in a more comprehensive way.
Comments 13: Lines 63-78. Please include the main results of the study within these initial lines of the discussion.
Response 13: Thank you for your comment. We have revised the initial lines of the discussion (page16, discussion, line67-70) to include a summary of the main results of the study. This addition emphasizes the original purpose of the study and improves clarity of the paper before delving further into discussion.
Comments 14: Discussion. In my opinion the discussion should be rewritten as its content supposes mainly a repetition of results, even with continuous references to tables and figures.
Response 14: We greatly appreciate your feedback. We have thoroughly revised the discussion section to reduce the redundancy of the mentions of results, especially from paragraph 2 and 4. Continuous references to tables and figures were deleted, except references to supplementary tables that were not mentioned in the result part. In addition to eliminating repetition, we separated the conclusion part (page18, conclusion section, line186) and supplemented it (page19, conclusion, line190-198) with information on how to apply our research, in order to further reorganize it into necessary content. In addition, we felt that the mention of physical activity and health was insufficient among the explanations of prior research, so we have included descriptions of studies on physical activity and dietary behaviors (page16, 17, paragraph 1, line72-73, 77-85).
Comments 15: The ethics committee approval and guidelines followed should be reported.
Response 15: Thank you for bringing this to our attention. We have included statements about the ethics committee approval and guidelines followed under the Institutional Review Board Statement section (page19, line215-219). Specifically, we have included the following: “All procedures were performed in accordance with the relevant guidelines and regulations of the Declaration of Helsinki. This study was an analysis of the publicly available data; thus, it did not require ethical approval. The data used in this study came from the KNHANES, and it has been annually reviewed and approved by the Korea Centers for Disease Control (KCDC) Research Ethics Review Committee since 2007.”
We hope that the revisions and additional details provided in this manuscript address your concerns adequately. We believe that these changes have significantly improved the quality of our paper.
Thank you for considering our revised manuscript and for your valuable feedback. We look forward to your positive response.
Reviewer 2 Report
Comments and Suggestions for Authors
General comments
The main aim of the presented study was to provide a more detailed understanding of how different aspects of physical activity, in-68 cluding intensity, duration, type, and purpose, may influence PA, an objective indicator 69 of better cellular health, in Korean adults after stratification by sex. Generally, the paper is well-written, easy to follow and adds merit to the scientific area of interests. However, I suggest some major improvements I presented below. Most of them are methodological flaws. All should be addressed before considering this work to be published.
Title
Title is well prepared.
Aim of the work
While the title clearly indicates what the authors wanted to explore and explain (Associations between Physical Activity and Phase Angle ...) the purpose is very poorly defined. Mainly, is very general (what does it mean: "comprehensive understanding"). Furthermore, I find inconsistencies between the purpose in the three main parts of the paper: the abstract, the body of the paper and the discussion. The wording of the objective needs to be standardised.
abstract:
provide a comprehensive understanding of how various aspects of physical activity, including intensity, duration, type, and purpose, may affect phase angle (PA), an objective indicator of better cellular health, in Korean adults after stratification by 14 sex.
main:
provide a more detailed understanding of how different aspects of physical activity, including intensity, duration, type, and purpose, may influence PA, an objective indicator 69 of better cellular health, in Korean adults after stratification by sex[
discussion:
identify whether physical activity is associated with high PA, an indicator of better cellular health, among Korean adults.
??? coprehensive or detailed (sic!), where cellular health suddenly came from
Abstract
The wording of the objective should agree with the other parts. The conclusion should indicate some application of the results related to the phase angle. There should be some measurable biophysical aspect.
Introduction
The wording of the objective should agree with the other parts.
Material and Methods
Participants
The sample size calculation in the context of statistical power is missing. Why 3996 participants were recruited? Is this the right number or will it cause an underestimation or overestimation of statistical significance. Moreover, what was the way of recruiting the participants (randomized or intended)? If randomized, what was method of randomization and sampling frame? I suggest making a flowchart to show the design and recruitment process for the study. This would be clearer for the reader.
Measures
The accuracy and validity of the InBody 970 should be presented (even if quoted). What is the reliability of the device compared to DEXA reference studies?
Physical activity
It is unclear whether the respondents completed a questionnaire (e.g. IPAQ) or whether the authors conducted the questions at their own discretion. What is the reliability of their own questions. The authors should address this issue very conscientiously and thoroughly. This part of the description must reliably indicate the replicability of the study.
Statistical analysis
I have no comments apart from the use of multinomial logistic regression. However, this part needs to be clearly explained.
Multinomial logistic regression is appropriate for any situation where a limited number of outcome categories (more than two) are being modeled and where those outcome categories have no order. The key is "have no order". Authors presented method as: "We also conducted a multinomial logistic regression to investigate the association between physical activity and the two subgroups of the above-average PA split based on the median." For me, data for subgroups with lower, average and above average condition are ordinal scale. In this case, a multinomial logistic regression is not correct. If data have an order (ordinal outcomes), the appropriate modeling approach for these outcome types is ordinal logistic regression. In fact, there are numerous known ways to approach the inferential modeling of ordinal outcomes. However, the most commonly adopted approach: proportional odds logistic regression. Proportional odds models (sometimes known as constrained cumulative logistic models) are more attractive than other approaches because of their ease of interpretation but cannot be used blindly without important checking of underlying assumptions.
Maybe I am wrong with defining subgroups with lower, average and above average condition, and Authors have arguments for a different approach, but this should be clearly and thoroughly explained in the Statistical analysis text.
Below I present the references to better understand Regression Modeling:
Keith McNulty. Handbook of Regression Modeling in People Analytics. Chapman&Hall/CRC Press.
Results
The results should be matched with a possibly improved method.
Discussion
I suggest to expand the discussion a little bit. I very strongly suggest supplementing the discussion with other very timely articles, particularly related to the PA patterns. Below I present sample, adequate works:
https://doi.org/10.3390/nu15030608
https://doi.org/10.3390/nu15051230
https://doi.org/10.3390/ijerph19116562
doi: 10.1093/pubmed/fdi082
doi: 10.3390/nu12072016
Conclusions
I suggest to separate paragraph Conclusions. Conclusions should be matched with a possibly improved method. In my opinion, there should additionally be a conclusion of an applied nature.
Author Response
Comments 1: General comments
The main aim of the presented study was to provide a more detailed understanding of how different aspects of physical activity, in-68 cluding intensity, duration, type, and purpose, may influence PA, an objective indicator 69 of better cellular health, in Korean adults after stratification by sex. Generally, the paper is well-written, easy to follow and adds merit to the scientific area of interests. However, I suggest some major improvements I presented below. Most of them are methodological flaws. All should be addressed before considering this work to be published.
Response 1: Thank you very much for your kind words and for the opportunity to revise our manuscript based on your insightful comments. We greatly appreciate the time and effort you have invested in evaluating our work, and we believe your suggestions have significantly improved the quality of our paper. In particular, your suggestion to conduct proportional odds logistic regression in replacement of multinomial logistic regression has allowed us to fix a major methodological flaw and enhance the integrity of our paper.
Below, we address each of your comments in detail. We have carefully considered each comment and have revised the manuscript accordingly to improve its quality and clarity. Thank you again for your insightful feedback and for considering our revised manuscript. We sincerely hope that the revised manuscript meets your expectations.
Comments 2:
Title
Title is well prepared.
Aim of the work
While the title clearly indicates what the authors wanted to explore and explain (Associations between Physical Activity and Phase Angle ...) the purpose is very poorly defined. Mainly, is very general (what does it mean: "comprehensive understanding"). Furthermore, I find inconsistencies between the purpose in the three main parts of the paper: the abstract, the body of the paper and the discussion. The wording of the objective needs to be standardised.
abstract:
provide a comprehensive understanding of how various aspects of physical activity, including intensity, duration, type, and purpose, may affect phase angle (PA), an objective indicator of better cellular health, main:
provide a more detailed understanding of how different aspects of physical activity, including intensity, duration, type, and purpose, may influence PA, an objective indicator 69 of better cellular health, in Korean adults after stratification by sex[
discussion:
identify whether physical activity is associated with high PA, an indicator of better cellular health, among Korean adults.
??? coprehensive or detailed (sic!), where cellular health suddenly came from
Response 2: Thank you for your feedback. We have revised the aim of our study to standardize the wording and remove any inconsistencies. This clarifies to the readers the aim of our study and prevents any confusion that may occur with “detailed understanding” and “comprehensive understanding”. In replacement of these phrases, we have used “association” to specify the type of understanding that this study aims to identify. Specifically we have standardized the wording of the aim under abstract section (page1, abstract, line12-14) and introduction section (page2, paragraph3, line78-80): “The aim of this work was to examine the association of various aspects of physical activity, including intensity, duration, type, and purpose, with phase angle, an objective indicator of health, in Korean adults after stratification by sex.” For the discussion section (page16, discussion, line66-67), we have used the wording: “This study aimed to identify the association of various aspects of physical activity, including intensity, duration, type, and purpose, with phase angle.”
Your comment about cellular health has brought to our attention that mentioning cellular health in our aim may bring confusion to the readers. Hence, we have replaced it with “health”. We thought “health” was more appropriate because the introduction lays out the relationships between phase angle and various health-related factors to show that phase angle is capable of being an indicator of health.
Comments 3:
Abstract
The wording of the objective should agree with the other parts. The conclusion should indicate some application of the results related to the phase angle. There should be some measurable biophysical aspect.
Response 3: Thank you for highlighting this area. We have expanded the conclusion section (page1, abstract, line35-37)(page19, conclusion, line190-198) to include a detailed description of applications of this study's results. Specifically, we have included information on the possible uses of this study to provide personalized health assessments, specific recommendations for health programs, personalized exercise prescriptions, and rehabilitation plans, particularly benefiting females with more demanding exercise requirements. Additionally, we have discussed how the easy accessibility of BIA such as InBody allows South Koreans to conveniently perform health assessments by tracking phase angle, a measurable biophysical parameter, changes over time, so they can monitor and adjust their exercise routines.
Comments 4:
Introduction
The wording of the objective should agree with the other parts.
Response 4: Thank you for your feedback. We have revised the objective of our study to keep the wording consistent. We have standardized the wording of the objective under introduction section (page2, paragraph3, line78-80): “The aim of this work was to examine the association of various aspects of physical activity, including intensity, duration, type, and purpose, with phase angle, an objective indicator of health, in Korean adults after stratification by sex.”
Comments 5:
Material and Methods
Participants
The sample size calculation in the context of statistical power is missing. Why 3996 participants were recruited? Is this the right number or will it cause an underestimation or overestimation of statistical significance. Moreover, what was the way of recruiting the participants (randomized or intended)? If randomized, what was method of randomization and sampling frame? I suggest making a flowchart to show the design and recruitment process for the study. This would be clearer for the reader.
Response 5: We appreciate your feedback regarding the sample size. We have revised the study population section (page2, study population and data section, line90-93, Figure1) to include a detailed description of the recruitment process and the sampling process of participants. Specifically, we have discussed how the source of our data, Korean National Health and Nutrients Examination Survey, a publicly accessible national survey was obtained. The recruitment process was carried out by the Korea Disease Control and Prevention Agency in a randomized way through a multi-stage clustered probability sampling method. The sampling process is stratified based on geographic regions to ensure that all subgroups of the population are adequately represented.
Furthermore, we have included a flowchart to improve visualization and clarification of the recruitment of 3996 participants who took part in the final analyses. 3996 participants is the right number, as 6265 participants took part in the survey in 2022 but 2269 participants were removed from the study due to having missing data.
Comments 6:
Measures
The accuracy and validity of the InBody 970 should be presented (even if quoted). What is the reliability of the device compared to DEXA reference studies?
Response 6: Thank you for your comment. While there was no available information regarding the accuracy and the validity of the InBody 970, there has been research that shows that Inbody 970 can be a reliable method for assessing appendicular lean mass, fat-free mass, and percentage body fat by comparing it to DEXA. We believe this sufficiently supports InBody as a reliable tool for measuring health markers (page3, phase angle section, line111-115).
Comments 7:
Physical activity
It is unclear whether the respondents completed a questionnaire (e.g. IPAQ) or whether the authors conducted the questions at their own discretion. What is the reliability of their own questions. The authors should address this issue very conscientiously and thoroughly. This part of the description must reliably indicate the replicability of the study.
Response 7: Thank you for your comment. Global Physical Activity Questionnaire(GPAQ), a previously validated questionnaire similar to IPAQ, was used to determine the physical activity parameters. The source of our data, Korea National Health and Nutrition Examination Survey (KNHANES), used IPAQ until 2014, but since 2014, KNHANES have been using GPAQ developed by WHO to make up for the shortcomings of IPAQ.(reference: https://doi.org/10.1177/1757975919854301, https://doi.org/10.5763/kjsm.2020.38.1.1) To enhance clarification regarding this, we have included a description of the basis of physical activity parameters (page3, physical activity section, line130-131): “The level of physical activity was assessed using the Global Physical Activity Questionnaire”.
Comments 8:
Statistical analysis
I have no comments apart from the use of multinomial logistic regression. However, this part needs to be clearly explained.
Multinomial logistic regression is appropriate for any situation where a limited number of outcome categories (more than two) are being modeled and where those outcome categories have no order. The key is "have no order". Authors presented method as: "We also conducted a multinomial logistic regression to investigate the association between physical activity and the two subgroups of the above-average PA split based on the median." For me, data for subgroups with lower, average and above average condition are ordinal scale. In this case, a multinomial logistic regression is not correct. If data have an order (ordinal outcomes), the appropriate modeling approach for these outcome types is ordinal logistic regression. In fact, there are numerous known ways to approach the inferential modeling of ordinal outcomes. However, the most commonly adopted approach: proportional odds logistic regression. Proportional odds models (sometimes known as constrained cumulative logistic models) are more attractive than other approaches because of their ease of interpretation but cannot be used blindly without important checking of underlying assumptions.
Maybe I am wrong with defining subgroups with lower, average and above average condition, and Authors have arguments for a different approach, but this should be clearly and thoroughly explained in the Statistical analysis text.
Below I present the references to better understand Regression Modeling:
Keith McNulty. Handbook of Regression Modeling in People Analytics. Chapman&Hall/CRC Press.
Response 8: (page1, abstract, line19), (page5, statistical analysis section, line197-200), (page16, paragraph1, line44-59, Figure2), (page17, paragraph2, line99-101) Thank you for your suggestion. You are absolutely right. Proportional odds logistic regression is used when the outcome variable has ordered categories, while multinomial logistic regression is used when the outcome variable has multiple unordered categories. As phase angle is categorized in an ordinal scale (below average, above average bottom 50th percentile, and above average top 50th percentile), proportional odds logistic regression is the more appropriate modeling method than multiple logistic regression. Using proportional odds logistic regression, we are able to examine the odds of being in a higher category of phase angle, the outcome variable. We have replaced the multinomial logistic regression with a proportional odds logistic regression to give a more accurate analysis of the association between physical activity and phase angle divided into ordered categories. The replaced figure is as follows. (Figure 2)
Comments 9:
Results
The results should be matched with a possibly improved method.
Response 9: (page16, paragraph1, line44-59, Figure2) Thank you for your suggestion. We have updated the results to reflect the proportional odds logistic regression analysis. Specifically we have included the following: “Figure 2 displays the results of proportional odds logistic regression, representing the analysis of the association between physical activity and the 3 subgroups of the phase angle (below average, and 2 subgroups of above-average phase angle split based on the median). It shows a similar trend to the main results. Among males, there is a statistically significant association where being sufficiently active (aOR=1.833, 95% CI=1.409-2.384) or insufficiently active (aOR=1.535, 95% CI=1.075-2.193) increases the odds of higher phase angle categories compared to being inactive. For females, the odds ratio of 1.131 (95% CI: 0.844 - 1.515) indicates a non-significant trend towards higher odds of higher phase angle categories for individuals who are insufficiently active compared to inactive, though this effect is not statistically significant at the conventional 5% level. When comparing females who are sufficiently active to those who are inactive, the odds of having a higher phase angle are 1.380 times higher for those who are sufficiently active, compared to those who are inactive.”
Comments 10:
Discussion
I suggest to expand the discussion a little bit. I very strongly suggest supplementing the discussion with other very timely articles, particularly related to the PA patterns. Below I present sample, adequate works:
https://doi.org/10.3390/nu15030608
https://doi.org/10.3390/nu15051230
https://doi.org/10.3390/ijerph19116562
doi: 10.1093/pubmed/fdi082
doi: 10.3390/nu12072016
Response 10: Thank you for your help in expanding our discussion. We greatly appreciate your kind explanation and specific guidance with reference we can look up. Upon review, have mentioned the implications of studies 1-3 you suggested in the first paragraph of the discussion section (page16, discussion, line72-73, line 77-85). It contains information on the relationship between physical activity patterns and dietary behaviors, which are important elements of health, and their relationship with metabolic syndrome. In addition, a study on physical activity patterns in diabetics and non-diabetics, and a study on the relationship between joint temporal dietary and physical activity patterns and various health indicators were attached. Lastly, since the first study you proposed addressed differences in physical activity and dietary behaviors depending on gender differences, in the 4th paragraph of the discussion section (page18, discussion, line146-148), we suggested that future research should also investigate the differences in dietary behavior in order to explain the gender differences in our study.
Comments 11:
Conclusions
I suggest to separate paragraph Conclusions. Conclusions should be matched with a possibly improved method. In my opinion, there should additionally be a conclusion of an applied nature.
Response 11: Thank you for your suggestion. We separated the conclusion part into a completely different section (page18, conclusion, line186) from the discussion section. And the conclusions derived from the proportional odds logistic regression are included in the discussion section (page17, discussion, line99-101). We also added more applications (page19, conclusion, line190-198) of our studies such as providing specific suggestions for better health programmes, personal exercise prescriptions, rehabilitation programmes, using phase angle as a personalized health assessment biophysical parameter, and changing perception of people who think it is enough to only do occupational physical activities.
We hope that the revisions and additional details provided in this manuscript address your concerns adequately. We believe that these changes have significantly improved the quality of our paper.
Thank you again for considering our revised manuscript and for your valuable feedback. We look forward to your positive response.
Round 2
Reviewer 1 Report
Comments and Suggestions for Authors
In my opinion the manuscript has been largely improved. All my comments have been properly addressed.
Reviewer 2 Report
Comments and Suggestions for Authors
I recommend to publish the article in new, corrected version.